# The Associated Factors of Prolonged Screen Time and Using Electronic Devices before Sleep among Elderly People in Shaanxi Province of China: A Cross-Sectional Study

**DOI:** 10.3390/ijerph18137020

**Published:** 2021-06-30

**Authors:** Yaxuan Zhang, Jiwei Wang, Xinyuan Lu, Beibei Che, Jinming Yu

**Affiliations:** School of Public Health, Fudan University, Shanghai 200433, China; joycezhangyx@163.com (Y.Z.); jiweiwang@fudan.edu.cn (J.W.); 18211020007@fudan.edu.cn (X.L.); 18211020047@fudan.edu.cn (B.C.)

**Keywords:** screen time, elderly people, using electronic devices before sleep

## Abstract

This study aimed to investigate prolonged screen time and using electronic devices before sleep and their associated factors in elderly people in Shaanxi province of China. We conducted a cross-sectional study among 2647 elderly participants aged 60–88 years. Data were collected through questionnaires. Demographic characteristics, screen time, using electronic devices before sleep, health status, lifestyles, sleep quality, and other associated factors were investigated. Logistic regression analysis was used to analyze the relationship between the associated factors of screen time and using electronic devices before sleep. The crude odds ratio (cOR) and adjusted odds ratio (aOR) and their 95% confidence intervals (CI) were calculated. A total of 1784 subjects completed the questionnaire. There were 6.89% participants with prolonged screen time and 13.45% using electronic devices before sleep frequently. Prolonged screen time was associated with personal monthly income (aOR = 1.205, *p* = 0.001), number of household residents (aOR = 0.860, *p* = 0.010), rural residents (aOR = 0.617, *p* = 0.038), and regular drinkers (aOR = 2.889, *p* < 0.001). Using electronic devices before sleep was associated with being female (aOR = 0.657, *p* = 0.007), family monthly income (aOR = 0.866, *p* = 0.002), being an occasional drinker (aOR = 1.891, *p* = 0.005), and self-reported sleep quality (aOR = 1.593, *p* = 0.007). In conclusion, several factors related to screen time or using electronic devices before sleep were identified. Only being a drinker was a common associated factor for both screen time and using electronic devices before sleep.

## 1. Introduction

Globally, the population is aging rapidly. According to the *World Population Prospects 2019*, 1 out of every 11 people in the global population was 65 years of age or older (9%) in 2019, and by 2050 this proportion will increase to 1 out of every 6 people (16%) [1]. China is a populous country and population aging is also an important issue in China. The proportion of people aged 65 and over in China has been rising and reached 12.6% in 2019 [2].

In these past years, the Chinese mobile communication market has undergone tremendous growth [3]. The *2020 Statistical Bulletin of the Communications Industry* in China reported that the total number of 4G users nationwide reached 1.289 billion, accounting for 80.8% of the number of mobile phone users in 2020 [4]. The number of 5G users continued to increase and, as of the end of April 2021, the number of 5G mobile phone terminal connections reached 310 million [5].

The increase in people’s screen time may contribute to undesirable consequences. Screen time was found to be always associated with sedentary behavior in [6]. Sedentary behavior has been found to be a risk factor for negative physical and mental health outcomes in children and adults [7,8]. Excessive screen sedentary time may lead to users becoming overweight and obese [6]. Studies have observed that TV viewing increases exposure to high energy density food advertisements [9,10,11], which have been shown to affect food choices at other times of the day [12] and increase snacking [13]. Excessive use of smartphones could have a negative impact on the health of users [14,15]. For example, people who overuse smartphones might suffer from loneliness, anxiety, irregular eating habits, blurred vision, wrist pain, sleep disorders, and fatigue [16,17,18,19,20]. In the past, it was believed that prolonged screen time was often related to children and young people. However, with the progress of society, more and more elderly people have begun to use various electronic devices proficiently. Smartphone-based medical monitoring has been used for chronic disease management among middle-aged and elderly people [21,22]. There is research focused on how to help the elderly use smartphones and improve their smartphone experience [23].

According to the theoretical model proposed by Cain and Gradisar in 2010, the increase in the use of electronic media, especially in the bedroom before sleep, is related to sleep disturbance, which is associated with impaired daytime functioning [24]. In 2018, Thomée reviewed research on smartphone use in bed and found that it was related to late sleep, shorter sleep duration, longer sleep onset latency, insomnia or sleep difficulties, lower sleep quality or sleep efficiency, and reduced daytime function or fatigue [25]. In addition, smartphone use in bed also related to mental impairments, such as depression, anxiety, and stress symptoms [25].

Recent evidence has suggested that prolonged screen time might be associated with higher income, non-intact family, urban residence, being male, poor sleep quality, bad self-reported health status, and obesity [26,27,28]. Despite important work in this area, the majority of current studies focus on screen time and the use of electronic devices before sleep among children and adolescents. In the context of population aging, our study focused on prolonged screen time and the use of electronic devices before sleep in elderly people. This research might provide an important opportunity to advance the understanding of excessive electronic device use among the elderly.

## 2. Materials and Methods

### 2.1. Participants

Data were collected between May and June 2018 on elderly people aged ≥60 years old from seven cities (Ankang, Hanzhong, Xi’an, Weinan, Baoji, Yan’an, and Yulin) and one demonstration zone (Yangling) in Shaanxi Province, China. We used convenience sampling to recruit urban and rural participants from districts and counties. Exclusion criteria included subjects who could not complete the questionnaire, such as elderly people with significant hearing or visual or mobility impairment. Considering that many elderly people suffer from presbyopia, we provided the subjects with presbyopic glasses in the survey.

A total of 2647 elderly people were investigated, with 1784 of them completing the questionnaire. Data were collected anonymously. Written informed consent was obtained from each participant. The study was performed in accordance with the Helsinki Declaration and approved by the Ethics Committee for Medical Research, School of Public Health, Fudan University (IRB00002408, FWA00002399; approval number IRB#2019-04-0741).

### 2.2. Independent Variables and Dependent Variables

The independent variables in the self-administered questionnaire included age, sex, weight, height, number of household residents, marital status (unmarried and not cohabiting with boyfriend/girlfriend; married or remarried or cohabiting with boyfriend/girlfriend; divorced or widowed), educational level (primary school and below; middle/high school; college and above), personal monthly income (six categories ranging from “CNY 0” to “≥CNY 4000”), family monthly income (six categories ranging from “CNY 0” to “≥CNY 8000”), square dancing (five categories ranging from “0 time per week” to “≥5 times per week”), place of residence (urban, rural), self-reported health status (five categories ranging from “very bad” to “very good”), smoking (non-smoker, occasional smoker, regular smoker), alcohol drinking (non-drinker, occasional drinker, regular drinker), self-reported sleep quality (good, poor), and having difficulty initiating sleep for at least 30 min (five categories ranging from “never” to “≥5 times per week”). Body mass index (BMI) was calculated as weight/(height^2^) and categorized as lean (BMI < 18.5), normal (18.5 ≤ BMI ≤ 23.9), overweight (23.9 < BMI ≤ 27.9), and fat (BMI > 27.9). Household residents were defined as family members who live with the participants for more than half a year within one year. The definition of square dancing was a dance performed by residents in outdoor places such as squares for fitness purposes; such dances are usually organized spontaneously by the masses and the participants are mostly middle-aged and elderly people.

Dependent variables were the daily screen time (short, prolonged) and the frequency of using electronic products 30 min before going to bed at night (less frequently, frequently). Screen time was defined as the time that people used electronic products (mobile phone/computer/tablet/television, etc.). The definition of short screen time was daily screen time of no more than four hours, while prolonged screen time was defined as daily screen time greater than or equal to four hours. There was no uniform standard for prolonged screen time. We referred to a previous study that defined prolonged screen time as daily screen time of more than four hours [28].

Participants were asked how often they used their electronic devices 30 min before sleep at night (never, rarely, sometimes, often, almost every day). Subjects who answered “almost every day” were defined as using electronic products 30 min before going to bed at night frequently, while others were considered as using electronic products before going to bed less frequently. 

### 2.3. Data Analysis

Data entry was completed using Epidata 3.1. Statistical analysis was performed using SAS version 9.4 (SAS Institute Inc., Cary, NC, USA; licensed to Fudan University). Descriptive statistics were calculated to study the sociodemographic characteristics of the sample. Continuous variables were presented as means with standard deviation (SD) and categorical variables were presented as frequencies and percentages.

Significance levels were set at 0.05. To examine the association between the independent variables and the outcome variables, logistic regression was performed to calculate the crude odds ratio (cOR) and stepwise variable selection was used in the multivariable logistic regression to obtain the adjusted odds ratio (aOR). Coefficients and *p*-values of each variable were calculated. Odds ratios were reported with 95% confidence intervals (95% CIs). The variables of personal monthly income, family monthly income, square dancing, self-reported health status, and having difficulty initiating sleep for at least 30 min were treated as continuous variables.

## 3. Results

### 3.1. Subject Characteristics and cORs

A total of 1784 participants were enrolled in our study. The participant characteristics and cOR are shown in Table 1. The average age of the participants was 68.3 (5.99 SD) years and over half (58.97%) of them were female. About 6.89% of the elderly people demonstrated prolonged screen time and 13.45% used electronic devices before sleep frequently. On average, about three to four people lived with the subjects. More than half (53.36%) of the participants had a normal BMI, and 41.26% were overweight or obese. Most (87.72%) subjects were married or remarried or cohabiting with their boyfriend/girlfriend. Only a few people (5.33%) had a college degree or above, half of them (50.78%) had a primary school degree or below, and the others (43.89%) had a middle or high school degree. The personal monthly income and family monthly income of the subjects were generally low: only 14.85% had a personal monthly income of CNY 3000 (=USD 465) or more and 12.11% had a family monthly income of CNY 8000 (=USD 1240) or more. About 22.75% of the participants participated in square dances. Fewer people (31.89%) lived in rural areas. Most participants (86.55%) had a general, good, or very good self-reported health status. Fewer people were regular smokers (12.95%) or regular drinkers (13.40%). Most participants (82.23%) had good self-reported sleep quality. About 59.87% of the participants had difficulty initiating sleep.

In univariate analysis, the factors positively associated with prolonged screen time were 23.9 < BMI ≤ 27.9 (vs. 18.5 ≤ BMI ≤ 23.9, cOR = 1.501, *p* = 0.042), personal monthly income (cOR = 1.330, *p* < 0.001), family monthly income (cOR = 1.200, *p* = 0.002), regular smoking (cOR = 2.001, *p* = 0.003), and alcohol drinking (occasional drinking vs. non-drinking, cOR = 1.876, *p* = 0.039; regular drinking vs. non-drinking, cOR = 3.363, *p* < 0.001). The factors negatively associated with prolonged screen time were the number of household residents (cOR = 0.847, *p* = 0.003), being female (cOR = 0.620, *p* = 0.011), educational level (primary school and below vs. college and above, cOR = 0.229, *p* < 0.001; middle/high school vs. college and above, cOR = 0.401, *p* = 0.002), and living in rural areas (cOR = 0.581, *p* = 0.015).

The results of the univariate analysis revealed that regular smoking (cOR = 1.719, *p* = 0.003), occasional drinking (cOR = 2.027, *p* = 0.001), and poor self-reported sleep quality (cOR = 1.455, *p* = 0.025) were positively associated with using electronic devices before sleep frequently, whereas being female (cOR = 0.632, *p* = 0.001) and family monthly income (cOR = 0.884, *p* = 0.007) were negatively associated with using electronic devices before sleep frequently.

### 3.2. The Associated Factors of Screen Time and Using Electronic Devices before Sleep

The results of the multivariable logistic regression are shown in Table 2. In our study, participants with a higher personal monthly income exhibited prolonged screen time. Subjects with fewer people living together with them exhibited longer screen time. Urban residents demonstrated more screen time than rural residents. Regular drinkers demonstrated longer screen time than non-drinkers.

The associated factors of using electronic devices before sleep are shown in Table 3. Male participants used electronic devices before sleep more frequently than females. Subjects with lower family monthly income used electronic devices before sleep more frequently. Occasional drinkers used electronic devices before sleep more frequently than non-drinkers. Participants with poor sleep used electronic devices before sleep more frequently.

## 4. Discussion

In our study, about 6.89% of the elderly participants reported daily screen time greater than or equal to four hours and 13.45% used electronic devices before sleep almost every day. The results of multivariable logistic regression indicated that prolonged screen time was associated with higher personal monthly income, having fewer household residents, being an urban resident, and regular drinking. Using electronic devices before sleep was associated with being male, a lower family monthly income, occasional drinking, and poor sleep quality.

Our study indicated that subjects with a higher personal monthly income reported longer screen time. In the context of the great mass of previous studies on screen time focused on children and adolescents, personal monthly income was not applicable for these groups. However, there is still evidence that higher socioeconomic status had a significant association with screen time, watching TV, and computer use, according to a study in Iranian children and adolescents [28]. A study among Hong Kong Chinese adults reported that screen time increased with income level [27] and studies in rural China indicated that a higher income could have a direct impact on smartphone use and that the use of smartphones in turn enhanced incomes [29,30]. We found that fewer household residents were significantly associated with prolonged screen time. Similarly, a cross-sectional study in Zhejiang, China, reported that non-intact family was positively associated with high screen time [26]. Another study of Iranian children and adolescents indicated that, with an increasing number of children in the family, the OR of watching TV was reduced, which also meant that people living with more family members might demonstrate shorter screen times [28]. In our study, urban participants had longer screen times than rural participants. The study among children and adolescents in Iran also found that urban residence was significantly associated with screen time [28]. However, the study in Zhejiang, China, reported that the difference in screen time between students in urban and rural areas was not statistically significant [26]. The results of this study showed that drinkers were associated with prolonged screen time and using electronic devices before sleep. Few groups have taken drinking into consideration due to the fact that most of the studies focused on children and adolescents, with the background that alcohol is not available to them in most countries around the world. One interesting finding from the research in Zhejiang, China, was that drinking carbonated beverages ≥3 times every day was positively associated with high screen time [26]. In our study, male participants were more likely to use electronic devices before sleep than females. This finding is in accord with recent studies among children or adolescents in China [26], Korea [31], Iran [28], Japan [32], Spain [33], Australia [34], and Brazil [35]. The current study found that lower family monthly income was significantly associated with using electronic devices before sleep frequently. This finding seems to be consistent with a cross-sectional study in Canada which reported that parental income was a negative predictor of children’s screen time [36]. Our results showed that participants with poor sleep were more likely to use electronic devices before sleep. That was also supported by previous studies among Iranian children and adolescents [28], early adolescents in American [37], and Japanese children [32].

The cross-sectional study in Zhejiang, China, reported that bad self-reported health status was positively associated with high screen time [26]. However, we did not find an association between self-reported health status and prolonged screen time among the elderly participants in our study. Moreover, a significant association was found between obesity and increased time spent watching TV in a study in Iranian children and adolescents [28]. This relationship was not found in our study.

This study suggested that many elderly people in Shaanxi Province of China exhibited prolonged screen time or used electronic devices before sleep frequently. The local government should pay more attention to this group, since there is much potential harm from excessive electronic devices use. Based on our study, the vulnerable elderly population may include males, people with higher personal income, those with fewer household residents, urban residents, drinkers, and people with poor sleep quality.

There were some limitations in our study. Prior studies have noted that people’s screen time might be associated with the family environment, social environment, and physical environment. Some of those environmental factors, which have been found to have an association with screen time, were not included in our study. For instance, parental habits at home were reported as an important factor influencing children′s screen time in one study [38]. Children’s screen time was found to be the result of an interaction between child and parent factors and was highly influenced by parental attitudes in another study [39]. A study focused on mothers from disadvantaged neighborhoods found that neighborhood cohesion was associated with screen time amongst mothers with both younger and older children [40]. A study among Chinese young children reported that, in rural areas, screen time was positively associated with traffic and limited places for and methods of outdoor play, and it was negatively associated with the importance of academics and the need for company [41]. It was found that adolescents with active friends were more likely to be physically active and spend less time engaging in screen-based behaviors [42]. Furthermore, our study population was elderly people, many of whom might suffer from cataracts, presbyopia, eyesight degeneration, and other eye diseases. We offered presbyopic glasses for them and our participants were those who could read and finish the research questionnaire. So those people with difficulty in reading or writing or who could not reach our survey spots were not included in our study. This survey was conducted before the COVID-19 pandemic, so our results cannot represent the current situation.

Those environmental factors mentioned above are important issues for future research. We noticed that there were many similar and related factors for screen time in our study among elderly people and in previous studies among children and adolescents. Further research should be undertaken to investigate whether those factors are associated with screen time in all ages and whether the situation has changed in the context of the COVID-19 pandemic.

## 5. Conclusions

In this study, we found that there were some elderly people in Shaanxi Province of China who exhibited prolonged screen time (6.89%) or used electronic devices before sleep frequently (13.45%). Prolonged screen time was associated with higher personal monthly income, fewer household residents, being an urban resident, and regular drinking. Using electronic devices before sleep frequently was associated with being male, a lower family monthly income, occasional drinking, and poor sleep quality.

## Figures and Tables

**Table 1 ijerph-18-07020-t001:** Characteristics of participants and cORs.

	Screen Time					Using Electronic Devices before Sleep				
	Short (*N* = 1661)	Prolonged (*N* = 123)	Total (*N* = 1784)	cOR	*p*-Value	Less Frequently (*N* = 1544)	Frequently (*N* = 240)	Total (*N* = 1784)	cOR	*p*-Value
Age (years), (mean ± SD)	68.3 ± 6.01	68.8 ± 5.76	68.3 ± 5.99	1.014	0.362	68.3 ± 6.01	68.3 ± 5.84	68.3 ± 5.99	0.999	0.925
Number of household residents, (mean ± SD)	3.6 ± 1.87	3.1 ± 1.82	3.6 ± 1.87	0.847	0.003 **	3.6 ± 1.85	3.6 ± 1.98	3.6 ± 1.87	0.996	0.918
Sex, n (%)										
Male	668 (91.26)	64 (8.74)	732 (41.03)	1.000	Reference	610 (83.33)	122 (16.67)	732 (41.03)	1.000	Reference
Female	993 (94.39)	59 (5.61)	1052 (58.97)	0.620	0.011 *	934 (88.78)	118 (11.22)	1052 (58.97)	0.632	0.001 **
BMI, n (%)										
BMI < 18.5	90 (93.75)	6 (6.25)	96 (5.38)	1.047	0.918	83 (86.46)	13 (13.54)	96 (5.38)	0.990	0.975
18.5 ≤ BMI ≤ 23.9	895 (94.01)	57 (5.99)	952 (53.36)	1.000	Reference	822 (86.34)	130 (13.66)	952 (53.36)	1.000	Reference
23.9 < BMI ≤ 27.9	544 (91.28)	52 (8.72)	596 (33.41)	1.501	0.042 *	522 (87.58)	74 (12.42)	596 (33.41)	0.896	0.483
BMI > 27.9	132 (94.29)	8 (5.71)	140 (7.85)	0.952	0.899	117 (83.57)	23 (16.43)	140 (7.85)	1.243	0.378
Marital status, n (%)										
Unmarried and not cohabiting with boyfriend/girlfriend	24 (96.00)	1 (4.00)	25 (1.40)	0.557	0.568	24 (96.00)	1 (4.00)	25 (1.40)	0.263	0.192
Married or remarried or cohabiting with boyfriend/girlfriend	1456 (93.04)	109 (6.96)	1565 (87.72)	1.000	Reference	1351 (86.33)	214 (13.67)	1565 (87.72)	1.000	Reference
Divorced or widowed	181 (93.30)	13 (6.70)	194 (10.87)	0.959	0.892	169 (87.11)	25 (12.89)	194 (10.87)	0.934	0.763
Educational level, n (%)										
Primary school and below	863 (95.25)	43 (4.75)	906 (50.78)	0.229	<0.001 ***	789 (87.09)	117 (12.91)	906 (50.78)	0.935	0.832
Middle/high school	720 (91.95)	63 (8.05)	783 (43.89)	0.401	0.002 **	673 (85.95)	110 (14.05)	783 (43.89)	1.031	0.923
College and above	78 (82.11)	17 (17.89)	95 (5.33)	1.000	Reference	82 (86.32)	13 (13.68)	95 (5.33)	1.000	Reference
Personal monthly income, n (%)				1.330	<0.001 ***				0.946	0.210
0	655 (95.62)	30 (4.38)	685 (38.40)			584 (85.26)	101 (14.74)	685 (38.40)		
CNY 1–999	336 (96.00)	14 (4.00)	350 (19.62)			297 (84.86)	53 (15.14)	350 (19.62)		
CNY 1000–1999	215 (93.48)	15 (6.52)	230 (12.89)			209 (90.87)	21 (9.13)	230 (12.89)		
CNY 2000–2999	226 (88.98)	28 (11.02)	254 (14.24)			224 (88.19)	30 (11.81)	254 (14.24)		
CNY 3000–3999	121 (82.88)	25 (17.12)	146 (8.18)			127 (86.99)	19 (13.01)	146 (8.18)		
≥CNY 4000	108 (90.76)	11 (9.24)	119 (6.67)			103 (86.55)	16 (13.45)	119 (6.67)		
Family monthly income, n (%)				1.200	0.002 **				0.884	0.007 **
CNY 0	267 (94.01)	17 (5.99)	284 (15.92)			242 (85.21)	42 (14.79)	284 (15.92)		
CNY 1–1999	337 (96.29)	13 (3.71)	350 (19.62)			297 (84.86)	53 (15.14)	350 (19.62)		
CNY 2000–3999	410 (93.82)	27 (6.18)	437 (24.50)			375 (85.81)	62 (14.19)	437 (24.50)		
CNY 4000–5999	295 (91.90)	26 (8.10)	321 (17.99)			272 (84.74)	49 (15.26)	321 (17.99)		
CNY 6000–7999	157 (89.20)	19 (10.80)	176 (9.87)			155 (88.07)	21 (11.93)	176 (9.87)		
≥CNY 8000	195 (90.28)	21 (9.72)	216 (12.11)			203 (93.98)	13 (6.02)	216 (12.11)		
Square dancing, n (%)				1.033	0.707				1.057	0.377
0 times per week	1283 (93.11)	95 (6.89)	1378 (77.24)			1186 (86.07)	192 (13.93)	1378 (77.24)		
Occasionally	147 (93.04)	11 (6.96)	158 (8.86)			148 (93.67)	10 (6.33)	158 (8.86)		
1–2 times per week	112 (94.92)	6 (5.08)	118 (6.61)			105 (88.98)	13 (11.02)	118 (6.61)		
3–4 times per week	44 (93.62)	3 (6.38)	47 (2.63)			41 (87.23)	6 (12.77)	47 (2.63)		
≥5 times per week	75 (90.36)	8 (9.64)	83 (4.65)			64 (77.11)	19 (22.89)	83 (4.65)		
Place of residence, n (%)										
Urban	1119 (92.10)	96 (7.90)	1215 (68.11)	1.000	Reference	1062 (87.41)	153 (12.59)	1215 (68.11)	1.000	Reference
Rural	542 (95.25)	27 (4.75)	569 (31.89)	0.581	0.015 *	482 (84.71)	87 (15.29)	569 (31.89)	1.253	0.120
Self-reported health status, n (%)				1.202	0.097				0.856	0.055
Very bad	28 (96.55)	1 (3.45)	29 (1.63)			21 (72.41)	8 (27.59)	29 (1.63)		
Bad	195 (92.42)	16 (7.58)	211 (11.83)			178 (84.36)	33 (15.64)	211 (11.83)		
General	743 (94.77)	41 (5.23)	784 (43.95)			685 (87.37)	99 (12.63)	784 (43.95)		
Good	562 (91.38)	53 (8.62)	615 (34.47)			526 (85.53)	89 (14.47)	615 (34.47)		
Very good	133 (91.72)	12 (8.28)	145 (8.13)			134 (92.41)	11 (7.59)	145 (8.13)		
Smoking, n (%)										
Non-smoker	1300 (93.80)	86 (6.20)	1386 (77.69)	1.000	Reference	1215 (87.66)	171 (12.34)	1386 (77.69)	1.000	Reference
Occasional smoker	157 (94.01)	10 (5.99)	167 (9.36)	0.963	0.913	143 (85.63)	24 (14.37)	167 (9.36)	1.192	0.454
Regular smoker	204 (88.31)	27 (11.69)	231 (12.95)	2.001	0.003 **	186 (80.52)	45 (19.48)	231 (12.95)	1.719	0.003 **
Alcohol drinking, n (%)										
Non-drinker	1322 (94.84)	72 (5.16)	1394 (78.14)	1.000	Reference	1225 (87.88)	169 (12.12)	1394 (78.14)	1.000	Reference
Occasional drinker	137 (90.73)	14 (9.27)	151 (8.46)	1.876	0.039 *	118 (78.15)	33 (21.85)	151 (8.46)	2.027	0.001 **
Regular drinker	202 (84.52)	37 (15.48)	239 (13.40)	3.363	<0.001 ***	201 (84.10)	38 (15.90)	239 (13.40)	1.370	0.106
Self-reported sleep quality, n (%)										
Good	1365 (93.05)	102 (6.95)	1467 (82.23)	1.000	Reference	1282 (87.39)	185 (12.61)	1467 (82.23)	1.000	Reference
Poor	296 (93.38)	21 (6.62)	317 (17.77)	0.949	0.834	262 (82.65)	55 (17.35)	317 (17.77)	1.455	0.025 *
Difficulty initiating sleep, n (%)				0.986	0.864				1.050	0.402
Never	667 (93.16)	49 (6.84)	716 (40.13)			612 (85.47)	104 (14.53)	716 (40.13)		
1–3 times per month	491 (92.99)	37 (7.01)	528 (29.60)			469 (88.83)	59 (11.17)	528 (29.60)		
1–2 times per week	286 (93.16)	21 (6.84)	307 (17.21)			269 (87.62)	38 (12.38)	307 (17.21)		
3–4 times per week	117 (91.41)	11 (8.59)	128 (7.17)			113 (88.28)	15 (11.72)	128 (7.17)		
≥5 times per week	100 (95.24)	5 (4.76)	105 (5.89)			81 (77.14)	24 (22.86)	105 (5.89)		

n, number; cOR was analyzed with one-way logistic regression; *: <0.05, **: <0.01, ***: <0.001.

**Table 2 ijerph-18-07020-t002:** Associated factors of prolonged screen time.

Associated Factors	Coefficient	*p*-Value	aOR	95% CI
Personal monthly income	0.187	0.001 **	1.205	(1.076, 1.350)
Number of household residents	−0.150	0.010 *	0.860	(0.767, 0.965)
Place of residence				
Urban (reference)				
Rural	−0.484	0.038 *	0.617	(0.391, 0.973)
Alcohol drinking				
Non-drinkers (reference)				
Occasional drinkers	0.430	0.171	1.536	(0.831, 2.842)
Regular drinkers	1.061	<0.001 ***	2.889	(1.859, 4.490)

*: <0.05, **: <0.01, ***: <0.001.

**Table 3 ijerph-18-07020-t003:** Associated factors of using electronic devices before sleep.

Associated Factors	Coefficient	*p* Value	aOR	95% CI
Sex				
Male (reference)				
Female	−0.420	0.007 **	0.657	(0.484, 0.892)
Family monthly income	−0.144	0.002 **	0.866	(0.790, 0.949)
Alcohol drinking				
Non-drinkers (reference)				
Occasional drinkers	0.637	0.005 **	1.891	(1.210, 2.954)
Regular drinkers	0.221	0.292	1.247	(0.827, 1.880)
Self-reported sleep quality				

**: <0.01.

## Data Availability

The datasets are available on reasonable request. If anyone wants to get access to the data, please send an email to joycezhangyx@163.com.

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
