# Peer review of "The Associated Factors of Prolonged Screen Time and Using Electronic Devices before Sleep among Elderly People in Shaanxi Province of China: A Cross-Sectional Study"

_ijerph, 2021, doi:10.3390/ijerph18137020_

Round 1

Reviewer 1 Report

This paper is interesting because it deals with elderly population in regard to screen time. I enjoyed reading it, still I think there are several spots that could be improved.

First, the research question is not clear yet. Is analysis searching for the determinants of two dependent variable (hours of screen time and frequency of bed-side device use)? Or do authors want to know about the general utility of device? If the point of the paper is about the formal, the paper needs more detailed hypothesis and literature review dealing with the factors that is included in the model.

Second, please define what electronic devise means in this study? does it include TV watching? or only smart device (cell phone, PDP device, smart device)? Or does it include computer (e.g., internet search or games) utility time.

Also, it would be nice if authors can provide the type of device and the utility hours for each device. It seems that TV would be the main driver of prolonged screen time in this case because the data tried to focus on general audience, rather than looking specifically elderly who use smart phone for own cell phone device.  Still the study measured total screen time. It means smart device use or computer use time could add on to the TV watching hours, and those smart devices use maybe attribute to the purchase power of the household.

Third, more detailed explanation is needed about the association between the income and screen time. Ans I think authors may able to find them paper abroad, like papers with dealing adult population or elderly population conducted in other Asian countries.  

Fourth, discussion part needs more improvements, and the policy implement can be addressed based on the results.

Reviewer 2 Report

The comments are in the attached document. 
